# Quantitative Ultrasound and Bone Health in Elderly People, a Systematic Review

**DOI:** 10.3390/biomedicines11041175

**Published:** 2023-04-13

**Authors:** Isabel Escobio-Prieto, María Blanco-Díaz, Elena Pinero-Pinto, Alvaro Manuel Rodriguez-Rodriguez, Francisco Javier Ruiz-Dorantes, Manuel Albornoz-Cabello

**Affiliations:** 1Department of Physiotherapy, Faculty of Nursing, Physiotherapy and Podiatry, University of Seville, 41009 Seville, Spain; iescobio@us.es (I.E.-P.); malbornoz@us.es (M.A.-C.); 2Institute of Biomedicine of Seville (IBIS), 41002 Seville, Spain; 3Faculty of Medicine and Health Sciences, University of Oviedo, 33006 Oviedo, Spain; blancomaria@uniovi.es; 4Physiotherapy and Translational Research Group (FINTRA-RG), Institute of Health Research of the Principality of Asturias (ISPA), Faculty of Medicine and Health Sciences, University of Oviedo, 33006 Oviedo, Spain; 5Doctoral Program in Health Sciences, University of Seville, 41004 Seville, Spain; fruiz13@us.es

**Keywords:** elderly, bone density, bone health, COVID-19, musculoskeletal health, QUS, DXA, chronic disease

## Abstract

Reduced bone mineral density (BMD), osteoporosis, and their associated fractures are one of the main musculoskeletal disorders of the elderly. Quickness in diagnosis could prevent associated complications in these people. This study aimed to perform a systematic review (SR) to analyze and synthesize current research on whether a calcaneal quantitative ultrasound (QUS) can estimate BMD and predict fracture risk in elderly people compared to dual-energy x-ray absorptiometry (DXA), following the PRISMA guidelines. A search was conducted in the main open-access health science databases: PubMed and Web of Science (WOS). DXA is the gold standard for the diagnosis of osteoporosis. Despite controversial results, it can be concluded that the calcaneal QUS tool may be a promising method to evaluate BMD in elderly people, facilitating its prevention and diagnosis. However, further studies are needed to validate the use of calcaneal QUS.

## 1. Introduction

Ageing leads to an increased risk of fractures [1,2,3,4] due to a decrease in bone strength, quality, and mass, which is called osteoporosis [5]. Osteoporosis produces an increase in frailty and in the risk of falling [5], with a consequent decrease in quality of life and high health expenses [1,3]. Prevention is the best strategy for osteoporosis, which should include identifying causative risk factors and early diagnosis [6].

Bone mineral density (BMD), in both men and women, is an independent risk of factor for fragility fractures, and also predicts the risk of fracture in people with osteopenia [7]. The change of BMD is a variable to take into account to predict the risk of fractures due to osteoporosis in the elderly [7].

DXA is the gold standard for the diagnosis of osteoporosis [8,9,10,11]. According to So E. et al. (2022), QUS has proved to be a promising tool for measuring BMD in patients with ankle fractures [8], while DXA devices have a relatively high cost and large size, which added to the limitation for people living in underdeveloped countries or rural environments. Moayyeri et al., in their 2012 meta-analysis, concluded that the accuracy in predicting bone fracture rates in older adult men and women is similar between Quantitative Bone Ultrasound (QUS) and DXA [12].

Nowadays, the use of more tools to define and estimate bone resistance and predict fracture risk is supported by imaging devices, fracture risk calculators, bone biopsy techniques, and laboratory tests [13].

Ultrasound is a mechanical wave that propagates in fluids and solid materials at frequencies higher than the human hearing range (about 20 kHz) [14]. It is suitable for probing bone biomechanical strength since the characteristics of its wave are closely related to the material and structural properties of the propagation medium [14].

The use of QUS was introduced in the field of osteoporosis in a study published in 1984 by Langton et al. [15]. Although it has been considered a potential method for the detection and diagnosis of osteoporosis in the elderly [9], QUS has less diagnostic accuracy than DXA, as it uses a simplified physical model of sound propagation in bone [16]. QUS has proven to be a promising non-invasive classification tool in the evaluation of osteoporosis [17].

Although the use of QUS in predicting BMD and bone strength in the elderly was first recorded over 30 years ago, the field has not yet reached maturity. The present review gathers the latest-published studies on the use of DXA versus calcaneal QUS with the aim of determining, synthesizing, and analyzing the use of the latter in terms of bone disease prevention and diagnosis, as well as its the effects in the elderly, as it is a non-invasive, economical, and portable tool that is also highly accepted by patients.

## 2. Materials and Methods

DXA is a method of diagnosis and initiation of treatment which measures the attenuation of X-rays in the mineral phase of the bone, and thus the calculation of BMD is performed [18].

QUS is a portable, relatively inexpensive, non-invasive detection tool that does not use ionizing radiation to determine the risk of fracture [18,19]. It also does not require professional technicians to carry it out [18,19]. Using QUS reduces the need for DXA [20,21]. This technology evaluates bone health by measuring the parameters related to the propagation of ultrasonic waves at different frequencies in the bone [19]. It provides information on bone structure, bone density [22], microarchitecture, and elasticity [23].

### 2.1. Search Strategy and Eligible Criteria

This systematic review (SR) followed the Preferred Reporting Items for Systematic Reviews and Meta Analyses (PRISMA) guidelines [24]. The protocol were registered in the International Prospective Register of Systematic Reviews (PROSPERO/NHS)-NUMBER CDR42022311302 [24,25]. Systematic research using PUBMed and Web of Science (WOS) was performed to identify trials suitable for inclusion in this SR. Keywords for the literature search were selected, with the authors’ agreement. Eligible criteria for the literature search for human studies were based on the PICO approach: (P—participants; I—interventions; C—comparison; O—outcomes) [25]. The terms used as keywords are listed in Appendix A, Table A1.

### 2.2. Study Selection

This SR screened studies based on the following exclusion criteria: articles that were not published in English or Spanish; studies conducted in young subjects, children, animals or people suffering from underlying associated pathologies; letters to the editor, case reports, poster presentations, narrative documents, SRs or non-systematic reviews and meta-analyses; and any document unrelated to the research problem. The bibliographic search focused on all articles published from 2012.

Two researchers (I.E.-P., M.A.-C.) independently reviewed the articles found (title and abstract screening, and full text reading). Articles that did not meet the inclusion criteria were discarded. A third author (E.P.-P.) intervened when the decision of these two researchers on the inclusion or exclusion of an article was split. Once the revision was completed, the search strategy was used again, in case new studies had been published, in order to analyze them and evaluate their inclusion. To formulate the objective and the research question, the PICO strategy was used [25] (P = elderly people; I = bone health diagnostic with QUS; C = control group, who received or did not receive treatment with DXA; O = variables related to clinical outcomes and health-related quality of life) (Table 1). This strategy enabled the establishment of critical reasoning [25] and the formulation of the following question: “What is the existing scientific evidence on the diagnosis of bone health in older adults through QUS versus DXA procedures?”.

After searching different keywords (Appendix A) in the aforementioned list of databases and sorting articles by title and summary, relevant articles were identified for complete reading, duplicate articles were eliminated, and the inclusion and exclusion criteria were applied to the sample of definitive data. Figure 1 shows the selection of the studies.

### 2.3. Data Sources

A systematic search of the following databases was performed for articles published before November 2022. Two researchers (I.E.-P., M.A.-C.) independently conducted an electronic literature search (up to date) on PubMed and WOS using the same methodology. Titles, abstracts, and full-text papers were screened and assessed to identify eligible articles, with M.B.-D, E.P.-P., A.O-P-V. and F.J.R.-D. intervening on split decisions between I.E.-P. and M.A.-C. To manage the data, we created a data summary sheet, based on Cochrane’s recommendations [26]. The data rescued from the studies were: Author and Year; Type of study; Number and type of patients; Type of interventions; Outcome measures; Primary results found among groups; and Conclusions and Limitations. The information was extracted and classified in the sections of interest for processing. Subsequently, the information provided by each article was compared. In this way, the results of the review work were obtained, which, compared to those provided by other authors, make up the conclusions of this study.

### 2.4. Methodological Quality Assessment and evaluation of the Risk of Bias in the Included Studies

The methodological quality and internal validity of the studies was carried out by two independent evaluators (E.P-P.; F.J.R.-D.), using the PEDro scale [27], which is based on the Delphy list developed by Verhagen et al. [28]. Studies with a score below 6 were considered to be low or level-1 evidence, while studies with a score of 6–8 were considered to be good evidence, and studies with 9–10 points were considered to be excellent evidence.

The risk of bias was considered and analyzed according to each study’s selection, detection, and reporting of bias, including randomization of the sample, allocation concealment, blinding of participants, blinding of outcome measures, loss of results, partial information of the results, and other biases. Appendix C, Table A3. This evaluation of the risk of bias and quality followed the directives of the Cochrane Handbook for Systematic Reviews of Interventions [25].

## 3. Results

### Selection of Studies

Applying the previously described search strategy, the following results were obtained in each database:PubMed: 21 articlesWOS: 28 articles

The flowchart (Figure 1) shows the screening that these articles were subjected to successively until the final sample was obtained, following the PRISMA criteria [24], and applying the inclusion and exclusion criteria described above.

In general, the publications included in this SR were published between 2012 and 2022, and they were focused on the use of DXA and QUS for the assessment of BMD and bone health in elderly people.

To establish the internal validity of the different studies, the articles were manually evaluated using the PEDro scale [26]. The different items, summarized in Appendix B, Table A2, were valued as 1 (1) or 0 (-) as a function of their respective presence or absence in the articles. The first item does not count in the total score, and thus the maximum score, including sections, was 10.

After manual evaluation of all articles, the score was verified with the PEDro database (http://www.pedro.org.au). In the case of disagreement, the evaluation was consulted with a second opinion to reach the final decision.

During the month of October 2022, the studies selected based on the criteria described in the methodology were analyzed. A total of 2066 patients from nine studies were analyzed to respond to the objectives of this SR. Table 2, shows the characteristics of the different studies: number and type of patients, age, type of intervention, outcome measures, variables, conclusions, and limitations.

Of all the articles, only one of them carried out a 20-year follow-up [28], using DXA and calcaneal QUS, two define the appropriate cut-off points for osteoporotic fractures with QUS [29], four of them perform measurements with DXA in spine, hip/femur, and with QUS in heel [30,31,32,33,34], one of them compares the results of 6 different QUS [30], and another study determines the number of DXAs that could be avoided with the use of QUS [35].

According to the included studies, the average sample was constituted by 174 participants, with a mean age of 75 years, ranging from 50 to 89 years. Overall, the participants were people with osteoporotic fractures, osteoporotic fracture risk, and postmenopausal women [21,28,29,30,31,32,33,34,35,36].

In every study, the patients were distributed equally and randomly into different groups, with no significant differences. Thus, three of the ten studies used a comparison control group of healthy elderly people [30,33,36].

**Table 2 biomedicines-11-01175-t002:** Characteristics of the studies selected.

Author	Type of Study	Participant Profile	Intervention	Variables	Results	Conclusions and Limitations
Chan M.Y., et al., 2012 [28]	EP	454 W 445 M 62–89 years	1989–2009 DXA QUS: BUA CUBA RX	DMO femoral neck: DXA CQUS (BUA): CUBA Fx for fragility: X-ray	75 M +154 W with fx for fragility W model BMD of the femoral neck and BUA had a higher AUC compared to the model without BUA Reclassification analysis showed: 7.3%, 11.1% and 5.2% for any fx.	CQUS: independent predictor of fx risk. QUS + BMD measurement could improve the predictive accuracy of fx risk in M AM.
Hadyi P., et al., 2015 [29]	ECC	205 W postMP 68–95 years (hip fx by OsP) 109 GC HM age 75.7 years	DXA 6 different QUS: Ach; Sh; IS; DMB; Omn; QUS-2	The outcomes of both groups were compared	T scores in W hip fx than matched controls: −2.38 vs. −1.64 (*p* < 0.001), −2.36 vs. −1.44 (*p* < 0.001) and −2.05 vs. −1.50 (*p* = 0.41). Ach, Sh, IS, and Omn QUS T-scores were also lower in W with hip fx compared to matched controls −3.20 vs. −2.36 (*p* < 0.001), −2.196 vs. −1.761 (*p* = 0.005), −2.631 vs. −1.849(0 < 0.001), −3.707 vs. −3.030 (*p* = 0.032). However, DBM and QUS-2 T-scores did not differ between groups. Compared with DXA (hip), the odds ratios of Ach, IS, and Sh were similar, while the odds ratios of DBM, Omn, and QUS-2 were significantly lower (*p* < =0.05).	Compared with DXA, Ach, Sh and IS can identify a clinically significant risk factor in W with high risk of hip FX.
Zha X.Y., et al., 2015 [31]	RCT	472 M over 60 years (78 years middle ages)	BMD in left hip and lumbar spine: DXA DMO CQUS Curve AUC	Evaluation of the OSTA/QUS; OSTA + QUS. Receiver operational characteristics analysis: SnD and EpD. AUC was compared.	-Prevalence of OsP 27.7% -Optimal cut for OSTA: −3.5 to predict M with OsP anywhere, with SnD= 47.3% and EpD= 76.8% The AUC for OSTA = 0.676 -Optimal cutting for QUS-T score: −1.25, with SnD = 80.4% and EpD = 59.7% AUC for QUS-T = 0.762 - QUS + OSTA Combination improved EpD = 92.9%, but reduced SnD = 36.1%.	OSTA and QUS, respectively, and OSTA + QUS may help find populations at high risk of OsP, which could be an alternative method for their diagnosis, especially in areas where DXA is not accessible.
Zhang L.C., et al., 2015 [30]	ETP	53 W OsP with femoral fx	CQUS DXA HSA	Surgery for femoral heads PCT Pearson correlation: QUS measured/DXA HSA Parameters Ym to evaluate the specific association: QUS (hip), femoral neck, trochanteric + Ward’s area, and femoral diaphysis.	Trochanteric area correlation coefficient (r = 0.356, *p* = 0.009) was >the neck area (r = 0.297, *p* = 0.031) and the total prox femur (r = 0.291, *p* = 0.034). QUS index was significantly correlated with HSA-derived parameters of the trochanteric area (r:0.315–0.356, all *p* < 0.05), as well as Ym of PCT of the femoral head (r = 0.589, *p* < 0.001).	The calcaneal bone and the trochanteric spongy bone showed a strong correlation. CQUS parameters may reflect the characteristics of the trochanteric area of the proximal hip, although it does NOT specifically reflect those of the neck or femoral shaft.
Cesme F., et al., 2016 [32]	ECC	20 M with fx hip 18 H with fx distal forearm 38 GC = age	DXA, DMO (CV and hip) CQUS	AUC to assess the discriminatory power of DXA FX and variable QUS	QUS T-score and SOS proved to be the best parameters for the identification of fx of hip and distal forearm. AUC is greater than DXA BMDs and other QUS parameters. QUS T of <= −1.18 could identify and rule out cases of hip FX approx. 80% of SnD and EpD, SOS <= 1529.75 reached almost 90% to rule out distal forearm fx.	The discrimination between M fx and non-fx with QUS variables was as good as DXA’s and even better.
Esmaeilzadeh S., et al., 2016 [36]	ECC	20 W with distal forearm fx and 18 M with hip fx 76 M = age as GC		DXA: measured BMD in CV, proximal femur and radius CQUS: measured bone acoustic parameters FRAX: calculated the probability of fx at 10 years. ORAI: in all participants ROC: evaluated the discriminatory power of fx of all tools	All variables’ probabilities demonstrated significant areas under the ROC curves for W discrimination with hip fx and those without fx. Only 33% of radium BMD, BUA attenuation, and FRAX.^®^ The highest probability of fx OsT calculated without BMD showed significant discriminatory power for distal forearm fx.	The QUS variables (BUA and FRAX^®^) are good candidates for the identification of both hip and distal radius fx.
Su Y., et al., 2018 [33]	ECC (Tree modeling study)	M&W > 65 years	QALY of the different OsP detection strategies, followed by a subsequent 5 years tto with alendronate/to no detection	DXA to all FRAX^®^ at specific thresholds QUS before DXA No screening	All screening strategies were systematically + cost-effective than the absence of detection in AM >65 years. One-way sensitivity analysis did not change results substantially. Probabilistic sensitivity analyses showed a dominant role of prescreening with FRAX followed by subsequent treatment with OsP drugs in people aged 70 years or +.	DXA-based OsP detection strategies with or without pre-detection (performed with FRAX QUS prior to DXA) are cost-effective + compared to the absence of detection in Chinese people over 65 years.
Fitzgerald G.E., et al., 2020 [21]	ETO	56 total. W post-MP 77% M > 50 58(7.2) years with axSpA	DXA: BMD of QoL and hip CQUS: BUA, SOS, SI and T scores. ROC analysis determined QUS’ ability to discriminate between low and normal BMD.	Calculate: nº of DXA that could be avoided. BASDAI BASFI ASQoL HAQ	BUA, SI and QUS T-score parameters correlated with BMD by DXA SOS did not All QUS parameters had the ability to discriminate between low and normal BMD (the area under the curve ranged from 0.695 to 0.779) QUS identified individuals without low BMD with 90% confidence, with BUA functioning better (SnD = 93%, negative predictive value = 86%).	Using QUS as a triage tool, up to 27% of DXA assessments could have been avoided. QUS could not confidently identify people with OsP. QUS is a promising NON-invasive classification tool in the evaluation of AM OsP with axSpa.
Li C.Z., et al., 2022 [35]	EO	82 p. > 50 years 12 M (62.3 ± 11.6 years) 70 W (63.9 ± 9.2 years)	BMD of the femoral and intertrochanteric neck of the left hip and lumbar spine (L1-L4) with DXA QUS parameters of the right and left calcaneus	DXA: lumbar spine+ left hip. BMD: T-scores; QUS (SONOS 3000):BQI +bilateral CQUS-T. (BQI = SOS and BUA) The mean value of both CQUS parameters. QUS T-score + BQI correlation of calcaneus + DXA parameters Lumbar spine and s.a. (software SPSS20.0) Was generated: receiver’s operating characteristic curve. Were evaluated: areas under the curves. Values for QUS were defined.	In M there was a moderate correlation between CQUS and prox femoral BMD (*p* < 0.05), no significant correlation between BMD of CQUS and lumbar BMD (*p* > 0.05). In W, CQUS were moderately correlated with BMD of the lumbar spine and prox femur (*p* < 0.05) DXA was used: Precision = 90.2%; SnD = 89.2%; EpD = 100%, Predictive value + =100%, Predictive value− =50%of CQUS in the diagnosis of OsP When the CQUS T-score was 1–1.8, the area under the curve =0.888, SnD = 73.21%, and EPD = 92.31% (*p* < 0.05). When the CQUS T-score was −2.35, SnD = 37.2%, and EpD = 100%.	QUS can be used to predict femoral BMD in middle-aged and elderly people, as well as lumbar to predict BMD in M. Calcaneal QUS has a good EpD as a screening method for OsP. It may be recommended for use as a pre-screening tool to reduce the number of DXAs. If the CQUS T score is −1.8, it has the highest diagnostic efficiency for OsP. When the CQUS T-score is <−2.35, it can be diagnosed as OsP.

PE: Prospective Study; RCT: Randomized clinical trial; ET: Cross-sectional study; EO, observational study; CCS: Case-Control Study; ETP: Prospective cross-sectional study; W: woman; M: men; HM: healthy mean; GC: control group; BUA, Broadband Ultrasound Attenuation; SOS, speed of sound; SI, stiffness index; QUI, quantitative ultrasound index; PED: elderly patients; postMP: postmenopausal; CT, computerized tomography; DXA, dual X-ray absorptiometry; QUS, quantitative ultrasound; CQUS: calcaneal QUS; FX: fractures; BMD, bone mineral density: OsP: Osteoporosis; AM: Senior; No.: number; HSA, Hip structural analysis; PCT: main compressive work; TTO: treatment; AUC: area under the curve; Ach: Achilles; Sh: Sahara; IS InSight; Omn: Omnisence; BASDAI: Bath AS disease activity index; Ym: young’s modulus; QALY: quality adjusted life years; BASFI: Bath AS functional index; ASQoL: AS Quality of Life; HAQ: Health Assessment Questionnaire; SnD: Sensitivity; EsD: Specificity; CV, spine; FRAX:^®^ fracture risk assessment tool; ORAI: Osteoporosis risk assessment instrument; p.: patients; ROC: analysis of receiver operating characteristics; axSpA: Spondyl axial arthropathy; s.a.: statistical analysis.

## 4. Discussion

With the increase in life expectancy in our society, maintaining a healthy lifestyle and improving the quality of life of elderly people is a goal for specialists in Geriatrics and Gerontology. It is known that fractures in the elderly represent a significant burden for health care and a decrease in quality of life [34]. Simple tools for risk assessment should be a priority option when it comes to quickly and inexpensively providing information about the elderly at risk of osteoporosis and fractures. Non-invasive diagnostic techniques such as calcaneal QUS are currently available to detect osteoporosis and predict fracture risk [8].

The present review, at a descriptive level, shows data consistent with those obtained in previous research regarding the use of calcaneal QUS versus DXA [33].

In the assessment of fracture risk, a large number of techniques are found [37], with DXA being the primary choice for treatment decisions. However, if DXA cannot be performed on BMD measurements, a fracture risk assessment can be carried out using clinical risk factors, and peripheral measurements, among several available tools. Some of the QUS technologies have demonstrated the potential to predict fracture risk [8] with DXA-like quality [38]. These tools, which have been prospectively validated, have obtained promising results and great advantages, such as being non-ionizing, cost-effective, and easily accessible for the evaluation of fracture risk, enabling an earlier treatment of the subject [8,38,39,40,41]. The difference in the use of each technique lies in the precision of the different tools; the method and the site of application will depend on the purpose of its use, such as diagnosis, fracture risk assessment, or follow-up of bone changes [42].

Although DXA is considered the “gold standard” for predicting osteoporotic fractures, QUS variables help to determine fracture risk [33,43]. Recent studies suggest that QUS is an independent predictor of fractures for both men and women, especially in low values of QUS [33,34,40]. These data are consistent with those of the meta-analysis conducted by McCloskey EV. et al., in 2015, in which they associated an increased risk of fractures (including hip fracture) with low QUS [33]. According to Li C.Z. et al. (2022), if the calcaneal QUS T score is −1.8, it has the highest diagnostic efficiency for osteoporosis. When the calcaneal QUS T-score is <−2.35, it can be diagnosed as osteoporosis [33]. Other studies demonstrate this ability to rule out osteoporosis with QUS as an alternative to DXA [8,12,42]. These findings disagree with those of Fitzgerald G.E. et al. (2020), who reported low prevalence in patients with osteoporosis, after recording the measurements with the QUS tool and failing to determine whether QUS plays a role in the discrimination of the participating subjects at high risk of osteoporosis [21].

The variables QUS, BUA, and FRAX^®^, which confirm a higher probability of osteoporotic fracture without BMD, can identify both radius and hip fractures [43]. Recent QUS approaches include guided waves to evaluate the mechanical and structural properties of long cortical bones, or to perform measurements not only in the calcaneus but also in the main sites of osteoporotic fracture, such as the hip and the spine [38], measured with DXA [21,33,38,44]. However, extensive DXA screening for population-wide osteoporosis is not recommended [33].

It is important to consider the incidence of fractures in people over 50 years of age, especially in those over 85 years of age, among whom osteoporosis is not adequately evaluated, and its prevalence is greater than the number of diagnoses recorded [36]. These data confirm the results of our systematic review [33]. The International Osteoporosis Foundation suggests that women over 65 years of age and men at risk of fractures get preventive screenings [45]. Su Y. et al. [46] reported that pre-screening with FRAX and pre-screening with QUS prior to DXA testing were consistently more effective and cost-effective for men 65 years and older and for women 70 years and older [46]. Screening in older people is generally more effective and prevents the use of DXA, which is consistent with other studies [21,35,36,46,47].

The results of the analyzed articles reveal significantly lower values in people with fractures compared to people without fractures, in both DXA and QUS [44], which is in line with other studies comparing DXA and QUS to identify low BMD or the likelihood of hip fracture [48], and lower limbs or osteoporosis with risk of fractures [49,50].

Ultrasound densitometry represents a novel proposal [21,33,36] in the clinical evaluation to determine the BMD of the subjects and to identify people at increased risk of fracture [51]. As musculoskeletal ultrasound is increasingly used in clinical practice, the need for innovation in its use has also increased [52].

Although the prediction of fractures of QUS is inferior to that of DXA [53], its advantages of portability, non-ionizing radiation, and low cost make this tool the most used [54], which is present in all the studies included in this review [21,28,29,30,31,32,33,35,36].

There are different calcaneal QUS devices available on the market, although not all of them are validated or obtain the same results in their measurements. While phalanx DBM and calcaneal QUS-2 could not discriminate between post-menopausal women with hip fracture and healthy control cases [29], other devices such as Achilles, Sahara, InSight, and Omni showed statistically significant data regarding hip fracture discrimination compared to the DXA reference tool [21,29,34]. These results are consistent with those of studies such as that of Zha X.Y. et al. (2015), who concluded that both OSTA and QUS and their combination can help to identify populations at high risk of osteoporosis [8]. On the other hand, the variables BUA and FRAX^®^ are good candidates in the identification of both hip and distal radius fractures [43]. In addition, the EPIDOS study [30], together with recent studies [29], underlines the importance of the SI of the Achilles device as the best short- and long-term predictor of hip fracture among QUS methods [55].

Research indicates that BUA predicts fractures in older women, and the association of BUA with fractures was similar to that of DXA [47]. In a recent SR [9], So E. et al., in patients with ankle fractures, confirmed that QUS is a promising tool for measuring BMD, and suggested that, for assessing BMD, these imaging techniques are secondary [9].

## 5. Conclusions

The method of measuring bone health in elderly people, using calcaneal QUS, can rule out a low BMD in elderly people, reducing the need for an evaluation with DXA. The literature grants validity and reliability to the calcaneal QUS method for the assessment of BMD in elderly people and for the prediction of the risk of fracture due to osteoporosis. If the T score of the calcaneal QUS is −1.8, it has the highest diagnostic efficacy for osteoporosis, while with a QUS T score <−2.35, osteoporosis can be diagnosed. Although DXA is the reference method for discriminating fracture risk and BMD in subjects, the use of QUS is reliable in the absence of DXA. Currently, for both techniques, new ways of evaluating bone quality and use are being developed. We can conclude that calcaneal QUS is a promising non-invasive classification tool for the evaluation of bone health in the elderly.

We believe that the findings provide an important exploration of the use and validity of QUS, as a predictor of fracture risk or low BMD in elderly people. While this is true, no quality ECAS has been found to analyze the tool. Further research should be conducted on the benefits of using QUS in healthy patients for preventive purposes and in subjects with possible decrease in AMD, osteoporosis, or who are at risk of suffering it and suffering a fall or fracture, as it is an easy-to-use, economical tool that does not require specialized training in its use. Therefore, it is considered that further studies are needed regarding the more widespread use of calcaneal QUS in clinical settings worldwide.

## Figures and Tables

**Figure 1 biomedicines-11-01175-f001:**
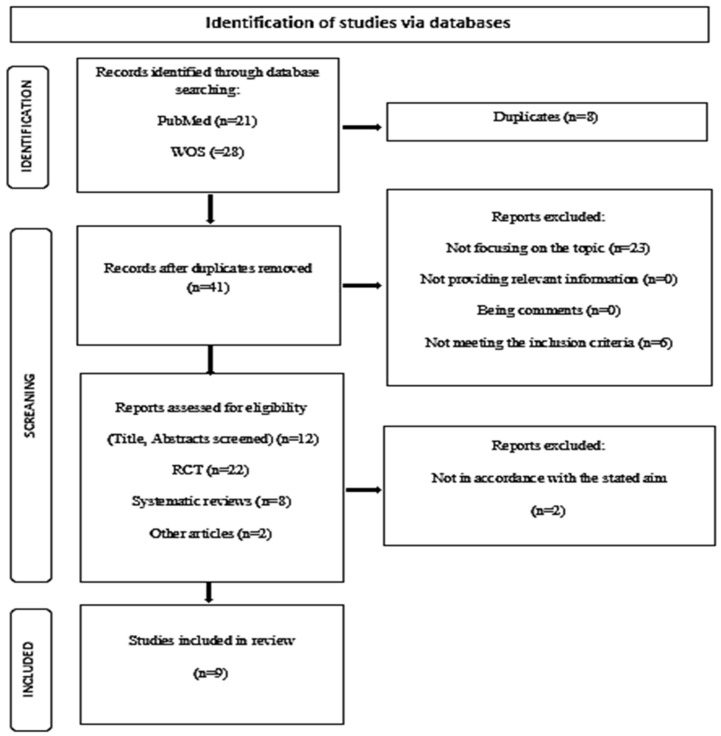
Flowchart showing the article selection. Developed by author following the guidelines of the PRISMA Guide 2021 [24].

**Table 1 biomedicines-11-01175-t001:** Measures used to assess results and effects.

Variables	
Primary Measures	Secondary Measures
- BMD: DXA in femoral neck and spine and calcaneal QUS - Risk of fracture due to osteoporosis	- FRAX^®^: probability of fracture in 10 years - ORAI: risk of osteoporosis - ROC: receiver operational characteristics analysis - QUI: quantitative ultrasound index

Developed by author.

## Data Availability

Not applicable.

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
