# Peer review of "Quantitative Ultrasound and Bone Health in Elderly People, a Systematic Review"

_biomedicines, 2023, doi:10.3390/biomedicines11041175_

Round 1

Reviewer 1 Report

The authors describe important findings comparing the accuracy and cost-effectiveness of QUS and DXA for predicting osteoporotic fractures. The methodology of this SR is appropriate and clearly explained.

Including the COVID pandemic in the article title and introduction seems unnecessary and distracting. The COVID issue is not carried through in the article. I suggest removing COVID and writing only about the comparison. There exists enough need for a DXA alternative in the absence of COVID.

Table 3: Replace "v" with "1".

Author Response

Response to Reviewer #1 Comments:

Ms. Ref. No.:  biomedicines-2313386

Title: Covid-19 indicates direct link to bone health: why wait? A Systematic Review on Quantitative Ultrasound and Bone Health in Elderly People”

REMARKS

COMMENTS

ACTIONS

The authors describe important findings comparing the accuracy and cost-effectiveness of QUS and DXA for predicting osteoporotic fractures. The methodology of this SR is appropriate and clearly explained.

Thank you very much for your comments, and for giving us the opportunity to correct the mistakes and justify our work.

Including the COVID pandemic in the article title and introduction seems unnecessary and distracting. The COVID issue is not carried through in the article. I suggest removing COVID and writing only about the comparison. There exists enough need for a DXA alternative in the absence of COVID.

We agree with the reviewer in this comment.

Apologies about this mistake.

The sentences have been revised.

Table 3: Replace "v" with "1".

We agree with the reviewer in the interest of this information. Apologies about this mistake.

The table has been revised, and fixed.

We hope all these changes can accomplish the reviewers’ and editor’s expectations about the manuscript’s quality improvement. Thanks again for all the comments and suggestions.

Reviewer 2 Report

This systematic review aims to explore the validity of calcaneal QUS in evaluating BMD in the aged population. The authors identified 10 articles with met the inclusion criteria and report that calcaneal QUS could be utilized as a predictor of fracture risk or low BMD in elderly people.

After careful consideration, this manuscript cannot be accepted for publication in this current form as major flaws can be observed. The authors need to improve the quality of the study and address the following points.

·       The link between COVID-19 and the purpose of the study is not clear and leads to confusion. Why do the authors try to associate the aim of the study with COVID-19? This is obviously evident by the inclusion of studies dated from 2012. All parts referring to COVID-19 have to be removed. If the authors wanted to link this, the bibliographic search should be focused on articles published from 2019.

·       The abstract needs to be rewritten as it does not effectively summarise the work.

·       Line 33, exposure to the sun – is there any evidence for that? A citation is needed.

·       Line 44, …etc – the information has to be included.

·       Line 92, which are the underlying associated pathologies?

·       The authors mention that other SRs have been excluded, however, in Appendix A it seems that SRs are included.

·       The number of the included articles (10) is too low for a reliable SR.

·       The study with the highest number of participants (Liu J.M., et al) is of the lowest quality based on PEDro scale criteria. Does this compromise the conclusions?

·       Tables 3 and 4 should be placed in Appendices and not in the main manuscript.

·       Based on international Organisations and numerous articles (e.g. WHO, OECD, PMID: 25580172), an “elderly” person is aged 65 or more. Here, the selected articles include individuals aged >40 years. How does this affect the conclusions?

·       In general, the variability of the interventions and variables of the included articles, based on Table 5, is very large. This needs to be narrowed in order for safe conclusions to be drawn. This is one of main flaws of the manuscript.

·       Since the authors state that “If the T score of the calcaneal QUS is -1.8, it has the highest diagnostic efficacy for osteoporosis, while with a QUS T score <-2.35, osteoporosis can be diagnosed.”, a meta-analysis with the appropriate participants (age), inclusion criteria etc (see all the above) and the corresponding Forest plots with subgroup analysis (based on calcaneal QUS only and not other anatomical sites) would support any conclusions. As it stands, the conclusions are very weak.

Author Response

Response to Reviewer #2 Comments:

Ms. Ref. No.:  biomedicines-2313386

Title: Covid-19 indicates direct link to bone health: why wait? A Systematic Review on Quantitative Ultrasound and Bone Health in Elderly People”

REMARKS

COMMENTS

ACTIONS

This systematic review aims to explore the validity of calcaneal QUS in evaluating BMD in the aged population. The authors identified 10 articles with met the inclusion criteria and report that calcaneal QUS could be utilized as a predictor of fracture risk or low BMD in elderly people.

After careful consideration, this manuscript cannot be accepted for publication in this current form as major flaws can be observed. The authors need to improve the quality of the study and address the following points.

Thank you very much for your comments, and for giving us the opportunity to correct the mistakes and justify our work.

·       The link between COVID-19 and the purpose of the study is not clear and leads to confusion. Why do the authors try to associate the aim of the study with COVID-19? This is obviously evident by the inclusion of studies dated from 2012. All parts referring to COVID-19 have to be removed. If the authors wanted to link this, the bibliographic search should be focused on articles published from 2019.

We agree with the reviewer in this comment.

Apologies about this mistake.

The sentences have been revised.

The abstract needs to be rewritten as it does not effectively summarise the work

We agree with the reviewer in the interest of this information. Apologies about this mistake.

The abstract has been revised, and fixed.

Line 33, exposure to the sun – is there any evidence for that? A citation is needed.

We agree with the reviewer in this comment.

Apologies about this mistake.

The introduction has been revised, and fixed.

Line 44, …etc – the information has to be included.

We agree with the reviewer in this comment.

Apologies about this mistake.

The introduction has been revised, and fixed.

Line 92, which are the underlying associated pathologies?

We agree with the reviewer in the interest of this information. We refer to other pathologies that may affect the state of the bone and that are not osteoporosis or osteopenia

The authors mention that other SRs have been excluded, however, in Appendix A it seems that SRs are included.

We agree with the reviewer in this comment.

Apologies about this mistake.

The Appendix A has been revised, and fixed.

The number of the included articles (10) is too low for a reliable SR.

Thank you for this information. It has consulted with the service of Bibliometrics of the University of Seville, and with the latest systematic reviews published in this journal (Biomedicines). And severan SR and Meta-Analyses have been published with 4/7 articles as a result. We cite two of them * (Pálinkás D., et al, 2023; Rossi J., et al., 2023).

In addition, as there are not too many articles the aim of the review is not to reach a conclusion about the effectiveness of the method in question, but to synthesize and analyze the existing evidence, inviting and guiding researchers to carry out new Clinical Studies.

The study with the highest number of participants (Liu J.M., et al) is of the lowest quality based on PEDro scale criteria. Does this compromise the conclusions?

Thank you very much for this appreciation. Actually, we thought a lot about whether to include this study, because of age. It was decided to include because the participants were people with osteoporotic fractures, osteoporotic fracture risk or postmenopausal women.

But, your comment has made us consider the elimination of this study.

We have decided to eliminate it because it is the one with the lowest score on the Pedro scale and because of the age of the participants.

Tables 3 and 4 should be placed in Appendices and not in the main manuscript.

We agree with the reviewer in this comment.

Apologies about this mistake.

The tables 2 and 4 have been placed in Appendices.

Based on international Organisations and numerous articles (e.g. WHO, OECD, PMID: 25580172), an “elderly” person is aged 65 or more. Here, the selected articles include individuals aged >40 years. How does this affect the conclusions?

We agree with the reviewer in this comment.

We have decided to eliminate it because it is the one with the lowest score on the Pedro scale and because of the age of the participants.

In general, the variability of the interventions and variables of the included articles, based on Table 5, is very large. This needs to be narrowed in order for safe conclusions to be drawn. This is one of main flaws of the manuscript.

Yes, we agree with the reviewer in this comment.

Apologies about this mistake.

The table 5 (now table 3) has been revised.

Since the authors state that “If the T score of the calcaneal QUS is -1.8, it has the highest diagnostic efficacy for osteoporosis, while with a QUS T score <-2.35, osteoporosis can be diagnosed.”, a meta-analysis with the appropriate participants (age), inclusion criteria etc (see all the above) and the corresponding Forest plots with subgroup analysis (based on calcaneal QUS only and not other anatomical sites) would support any conclusions. As it stands, the conclusions are very weak.

We agree with the reviewer in this comment. The intention of the authors is to carry out a synthesis and analysis of the existing scientific evidence to date, to help the investigation of clinical trials using QUS as a diagnostic method.

We hope all these changes can accomplish the reviewers’ and editor’s expectations about the manuscript’s quality improvement. Thanks again for all the comments and suggestions.

*SR Biomedicines (2023):

Pálinkás, D.; Teutsch, B.; Gagyi, E.B.; Engh, M.A.; Kalló, P.; Veres, D.S.; Földvári-Nagy, L.; Hosszúfalusi, N.; Hegyi, P.; ErÅ‘ss, B. No Association between Gastrointestinal Rebleeding and DOAC Therapy Resumption: A Systematic Review and Meta-Analysis. Biomedicines 202311, 554. https://doi.org/10.3390/biomedicines11020554

Rossi, J.; Cavallieri, F.; Bassi, M.C.; Biagini, G.; Rizzi, R.; Russo, M.; Bondavalli, M.; Iaccarino, C.; Pavesi, G.; Cozzi, S.; Giaccherini, L.; Najafi, M.; Pisanello, A.; Valzania, F. Efficacy and Tolerability of Perampanel in Brain Tumor-Related Epilepsy: A Systematic Review. Biomedicines 202311, 651. https://doi.org/10.3390/biomedicines11030651

Reviewer 3 Report

The present review  gathers latest published studies on the use of DXA versus calcaneal QUS with the aim of  determining the validity of the latter in terms of bone disease prevention and diagnosis,  as well as its the effects, in the elderly, as it is a non-invasive, economical and portable tool  that is also highly accepted by patients. In conclusion, the method of measuring bone health in elderly people, using calcaneal QUS, can  rule out a low BMD in elderly people with osteoporosis, reducing the need for an evaluation with DXA. The literature grants validity and reliability to the calcaneal QUS method  for the assessment of BMD in elderly people and for the prediction of the risk of fracture due to osteoporosis. If the T score of the calcaneal QUS is -1.8, it has the highest diagnostic  efficacy for osteoporosis, while with a QUS T score <-2.35, osteoporosis can be diagnosed.  Although DXA is the reference method for discriminating fracture risk and BMD in subects, the use of QUS is reliable in the absence of DXA.

The authors  have made a series of references to the relationship between bone and covid in the introduction. A striking fact that is not reported is that during the pandemic the number of hip fractures decreased. This introduction is not related to the development of the study, which consists of a systematic review of DEXA and QUS unrelated to the COVID pandemic.

The methodology is complete, widely described, which would allow the study to be carried out by another research group. The results are clearly expressed, defining the biases and the data of the studies clearly.

The discussion is correct, adapting to the results obtained

Author Response

Response to Reviewer #3 Comments:

Ms. Ref. No.:  biomedicines-2313386

Title: Covid-19 indicates direct link to bone health: why wait? A Systematic Review on Quantitative Ultrasound and Bone Health in Elderly People”

REMARKS

COMMENTS

ACTIONS

The authors have made a series of references to the relationship between bone and covid in the introduction. A striking fact that is not reported is that during the pandemic the number of hip fractures decreased. This introduction is not related to the development of the study, which consists of a systematic review of DEXA and QUS unrelated to the COVID pandemic

Thank you very much for your comments, and for giving us the opportunity to correct the mistakes and justify our work.

We agree with the reviewer in the interest of this information. Apologizes about this mistake.

The introduction has been revised.

The methodology is complete, widely described, which would allow the study to be carried out by another research group. 

 Thank you very much for your comments

The results are clearly expressed, defining the biases and the data of the studies clearly.

Thank you very much for your comments

The discussion is correct, adapting to the results obtained

Thank you very much for your comments

We hope all these changes can accomplish the reviewers’ and editor’s expectations about the manuscript’s quality improvement. Thanks again for all the comments and suggestions.

Round 2

Reviewer 1 Report

The authors have made the changes suggested. Minor corrections needed:

Line 19: "agility" is not an appropriate word here.

57-58: incomplete sentence

65: define DXA, QUS

304: "...rule out a low BMD in elderly people with osteoporosis..." This does not make sense because people with osteoporosis have low BMD.

Author Response

Response to Reviewer #1 Comments:

Ms. Ref. No.:  biomedicines-2313386

Title: Quantitative Ultrasound and Bone Health in Elderly People, a Systematic Review”

REMARKS

COMMENTS

ACTIONS

The authors have made the changes suggested.

Minor corrections needed:

Line 19: "agility" is not an appropriate word here.

Thank you very much for your comments, and for giving us the opportunity to correct the mistakes and justify our work.

We agree with the reviewer. Apologizes about this mistake.

The word “Agility” was eliminated and replaced by “Quickness”

57-58: incomplete sentence

 Thank you very much for your comments

The sentence has been revised.

65: define DXA, QUS

Thank you very much for your comments

Has been added

304: "...rule out a low BMD in elderly people with osteoporosis..." This does not make sense because people with osteoporosis have low BMD.

Thank you very much for your comments. (Lines 266/267)

The sentenced has been revised.

We hope all these changes can accomplish the reviewers’ and editor’s expectations about the manuscript’s quality improvement. Thanks again for all the comments and suggestions.

Reviewer 2 Report

The authors addressed my comments

Author Response

Thank you very much for your comments, and for giving us the opportunity to correct the mistakes and justify our work.

Reviewer 3 Report

The questions have been answered by the authors

Author Response

(The authors gave the same response as above.)
